# Functional and Kinetic Comparison of Alanine Cysteine Serine Transporters ASCT1 and ASCT2

**DOI:** 10.3390/biom12010113

**Published:** 2022-01-11

**Authors:** Jiali Wang, Yang Dong, Christof Grewer

**Affiliations:** Department of Chemistry, Binghamton University, 4400 Vestal Parkway East, Binghamton, NY 13902, USA; jwang204@binghamton.edu (J.W.); ydong45@binghamton.edu (Y.D.)

**Keywords:** neutral amino acid transporter, electrophysiology, kinetics, rapid solution exchange

## Abstract

Neutral amino acid transporters ASCT1 and ASCT2 are two SLC1 (solute carrier 1) family subtypes, which are specific for neutral amino acids. The other members of the SLC1 family are acidic amino acid transporters (EAATs 1–5). While the functional similarities and differences between the EAATs have been well studied, less is known about how the subtypes ASCT1 and 2 differ in kinetics and function. Here, by performing comprehensive electrophysiological analysis, we identified similarities and differences between these subtypes, as well as novel functional properties, such as apparent substrate affinities of the inward-facing conformation (in the range of 70 μM for L-serine as the substrate). Key findings were: ASCT1 has a higher apparent affinity for Na^+^, as well as a larger [Na^+^] dependence of substrate affinity compared to ASCT2. However, the general sequential Na^+^/substrate binding mechanism with at least one Na^+^ binding first, followed by amino acid substrate, followed by at least one more Na^+^ ion, appears to be conserved between the two subtypes. In addition, the first Na^+^ binding step, presumably to the Na3 site, occurs with high apparent affinity (<1 mM) in both transporters. In addition, ASCT1 and 2 show different substrate selectivities, where ASCT1 does not respond to extracellular glutamine. Finally, in both transporters, we measured rapid, capacitive charge movements upon application and removal of amino acid, due to rearrangement of the translocation equilibrium. This charge movement decays rapidly, with a time constant of 4–5 ms and recovers with a time constant in the 15 ms range after substrate removal. This places a lower limit on the turnover rate of amino acid exchange by these two transporters of 60–80 s^−1^.

## 1. Introduction

ASCTs (alanine serine cysteine transporters) belong to the SLC1 family of membrane transport proteins. Two subtypes of this family, ASCT1 and ASCT2 (SLC1A4 and SLC1A5) are neutral amino acid transporters, in contrast to the remaining five members of the family (excitatory amino acid transporters, EAATs), which transport acidic amino acids [1,2,3,4,5,6,7]. ASCT2 was previously shown to function as a Na^+^-dependent neutral amino acid exchanger, which requires the presence of Na^+^, but does not use the Na^+^ gradient as a driving force for amino acid uptake [2,3,4,5,8]. In this mechanism, extracellular amino acid is exchanged with intracellular amino acid, without the ability of the empty transporter to complete the transport cycle.

In previous studies, a high expression level of ASCT2 was observed in cancer cells, suggesting an association with cancer metabolism. It was suggested to directly or indirectly provide energy and nitrogen resources to rapidly growing cells, by taking up glutamine [9,10,11,12]. In addition, ASCT1 and 2 may be involved in neuronal disorders. For example, SLC1A4 is the pathogenic gene of a neurologic disorder manifesting in intellectual disability, postnatal microcephaly, spasticity and thin corpus callosum [13].

Within the SLC1 family, the various subtypes share a high sequence similarity and identity [14]. ASCT1 and 2 share 57% sequence identity and ASCTs share up to 29% sequence identity with EAATs [12,14]. The substrate binding site, however, shows distinct differences, accounting for the diverging substrate specificities of ASCTs and EAATs. The excitatory amino acid transporters have a positively-charged side chain (highly conserved arginine) in the binding site, which forms an ion pair with the negatively charged amino acid side chain [15,16,17]. In the ASCTs, this cationic side chain is replaced with a neutral amino acid, for example cysteine in ASCT2. In addition to diverging amino acid substrate specificities, EAATs and ASCTs also have significant differences in function. For example, EAATs do not function as exchangers and, in contrast to ASCTs, counter-transport K^+^ [18,19]. Even among the EAAT subtypes, functional differences exist. For example, EAATs 4–5 show slower kinetics and distinct voltage dependence of transport compared to EAATs 1–3 [20,21,22,23,24,25]. In addition, cation selectivities are different within the EAATs, in particular with respect to the ability to transport Li^+^ [18,26,27]. Therefore, it is possible that ASCT1 and 2 also show distinct functional properties, although little is known about the details of kinetics and voltage dependence of ASCT1 amino acid exchange.

While the structure of ASCT1 is not known, structures of ASCT2 have recently been published in both the substrate-bound inward- and outward-facing conformations, as well as in complex with a competitive inhibitor [28,29,30,31]. These structures highlight the importance of the hairpin loop 2 (HP2), which locks the substrate into the binding site and acts as both the internal and external gate in the elevator-like alternating access transport mechanism. However, the kinetic properties of the inward-facing amino acid binding site, for example the apparent *K*_m_ values for substrates, are not well characterized [32].

Another important topic is substrate selectivity. In a previous study, it was shown that, in contrast to ASCT2, ASCT1 does not recognize glutamine as a substrate [3]. However, apart from a review article published while this work was in preparation [14], no other reports have focused on the functional similarities and differences between ASCT1 and ASCT2. Here, we applied electrophysiological techniques to comprehensively study the kinetic properties of ASCT1 and 2. Our results demonstrate that ASCT1 has stronger apparent affinity for amino acid substrates alanine and serine than ASCT2. In addition, Na^+^ interacts more strongly with ASCT1 than with ASCT2. Finally, the leak anion conductance is much less pronounced in ASCT1 than ASCT2. Interestingly, ASCT1 and ASCT2 showed similarity in kinetic properties when probing the electrogenic steps associated with substrate translocation. Both transporters exhibited amino acid-induced rapid charge movement, which recovered upon substrate removal with a time constant in the 15 ms range, placing an upper limit on the ASCT1 and ASCT2 cycle time.

## 2. Materials and Methods

### 2.1. Cell Culture and Transfection

HEK293 cells (American Type Culture Collection No. CRL 1573) were cultured as described previously. Cell cultures were transiently transfected with wild-type or mutant hASCT1, hASCT2 cDNAs, inserted into a modified pBK-CMV-expression plasmid. Transfection were performed according Jetprime transfection reagent and protocol (Polyplus). The cells were transfected after 24–36 h then used for electrophysiological analysis.

### 2.2. Electrophysiology

Currents associated with alanine serine cysteine transporters were measured in the whole-cell current recording configuration. Whole-cell currents were recorded with an EPC7 patch-clamp Amplifier (ALA Scientific, Westbury, NY, USA) under voltage-clamp conditions. The resistance of the recording electrode was 3–6 MΩ. Series resistance was not compensated because of the small whole-cell currents carried by ASCTs. The composition of the solutions for measuring amino acid exchange currents in the anion conducting mode was: 140 mM NaMes (Mes = methanesulfonate), 2 mM MgGluconate_2_, 2 mM CaMes_2_, 10 mM 4-(2-Hydroxyethyl)piperazine-1-ethanesulfonic acid (HEPES), pH 7.3 (extracellular) and 130 mM NaSCN (SCN = thiocyanate), 10 mM Ser, 2 mM MgGluconate_2_, 5 mM Ethylene glycol-bis(2-aminoethylether)-*N*,*N*,*N*′,*N*′-tetraacetic acid (EGTA), 10 mM HEPES, pH 7.3 (intracellular), as published previously [19]. For the measurement of the transport component of the current, intracellular SCN^−^ was replaced with the Mes^-^ anion.

### 2.3. Voltage-Jump Experiments

Voltage jumps (−100 to +60 mV) were applied to perturb the translocation equilibrium and to determine the voltage dependence of the anion conductance. To determine ASCT-specific currents, external solution contained 140 mM NaMes in the presence of varying concentrations of amino acid substrate. The internal solution contained 130 mM NaSCN in the presence of 10 mM amino acid substrate. Competitive blocker (R)-gamma-(4-biphenylmethyl)-L-proline [33] was used in control voltage jump experiments, yielding the unspecific current component, which was subtracted from the total current. Capacitive transient compensation and series resistance compensation of up to 80% was employed using the EPC-7 amplifier. Non-specific transient currents were subtracted in Clampfit software (Molecular Devices).

### 2.4. Rapid Solution Exchange

Fast solution exchanges were performed using the SF-77B (Warner Instruments, LLC, Holliston, MA, USA) piezo-based solution exchanger, allowing a time resolution in the 10–20 ms range. Amino acid substrate was applied through a theta capillary glass tubing (TG200-4, OD = 2.00 mm, ID = 1.40 mm. Warner Instruments, LLC, MA, USA), with the tip of the theta tubing pulled to a diameter of 350 μm and positioned at 0.5 mm to the cell [34]. For paired-pulse experiments, currents were recorded with varying interval time after removal of amino acid, starting at 10 ms.

### 2.5. Data Analysis

The data analysis was performed in Microsoft Excel and Microcal Origin software. Error bars are shown as mean ± stand deviation, collected from recordings of 6 to 10 cells, for statistical analysis. To determine substrate and Na^+^ apparent *K*_m_ values, non-linear curve fitting was used with a Michaelis–Menten-like equation, I = I_max_ × [substrate]/(*K*_m_ + [substrate]), where I_max_ is the current at saturating substrate concentration.

Transient signals of piezo-based solution-exchange results were analyzed in Clampfit software (Axon Instruments) by fitting with a sum of two exponential components. I = I_1_·exp(−t/τ_rise_) + I_2_·exp(−t/τ_decay_). Here, I is the current amplitude, τ the time constant and t the time.

## 3. Results

### 3.1. ASCT1 Binds Amino Acid Substrate with Higher Apparent Affinity than ASCT2 in the Outward-Facing State

To test the function of ASCTs, including apparent substrate affinities, we selected the HEK293 cell expression system for transfection with ASCT cDNAs, together with electrophysiological analysis in the whole cell recording configuration. In this system, alanine did not elicit currents in non-transfected cells. Typical currents recorded from human ASCT2 transfected cells in response to extracellular solution exchange to various concentrations of serine are shown in Figure 1A. As expected, application of serine activated the ASCT2 anion conductance in the presence of the intracellular anion, SCN^−^, indicating the population of the substrate-induced anion-conducting state. No steady-state currents were observed in the absence of permeating anions and at zero-*trans* conditions for substrate and sodium, as discussed later (Section 3.5). Inward currents were concentration dependent and were analyzed with a Michaelis–Menten-like equation, to yield the apparent *K*_m_ for the substrate, serine (Figure 1B) as 350 ± 60 μM. *K*_m_ values for other amino acid substrates in comparison with ASCT1 are summarized in Figure 1C, indicating that substrate apparent affinities are 2–5 times higher in ASCT1 than in ASCT2. Consistent with previous results, L-glutamine did not generate currents in ASCT1-expressing cells. These data confirm previous results that neutral amino acids serine and alanine are major substrates for ASCTs but glutamine only binds to ASCT2.

### 3.2. Leak Anion Conductance Is Less Pronounced in ASCT1 than in ASCT2

ASCT anion conductance was tested using SCN^-^ as a highly permeable anion at the intracellular side of the membrane. As expected, application of amino acid substrate at a concentration of 500 μM generated inward current, which increased with negative membrane potential (Figure 2A, voltage jump protocol illustrated in the top panel). This inward current is caused by SCN^-^ leaving the cell. Steady-state current was then plotted as a function of the membrane voltage, as shown in the I-V curves in Figure 2B,C. Serine induced the largest anion current in both transporters, while glutamine was ineffective in generating anion current in ASCT1 at any voltage tested. Interestingly, ASCT1 showed a much smaller leak anion current, relative to the substrate-induced current, than ASCT2. This current was observed in the absence of amino acid substrate, but in the presence of Na^+^. It is specific for ASCTs, as background was subtracted using a competitive inhibitor, which blocks the leak anion conductance.

### 3.3. Intracellular Substrate Inhibits the Leak Anion Conductance in ASCT1

Next, we tested the substrate binding properties of the inward-facing state by applying amino acid from the intracellular side of the membrane. This was done using the inside-out patch configuration, allowing the application of intracellular substrate through solution exchange, as previously described for other transporters [35]. From earlier studies with the EAAT SLC1 family members, it is known that intracellular substrate application activates the anion conductance, but with a 20 times lower glutamate apparent affinity (intracellular glutamate apparent affinity is about 100 μM compared with 5 μM in the outward-facing conformation) [35]. As ionic conditions, we used SCN^-^ in the inside-out recording pipet, together with 140 mM NaMes and 10 mM serine. Thus, the amino acid binding site should be inward-facing before serine was applied to the inside-out patch (Figure 3). As shown in Figure 3C, serine activated the anion conductance in ASCT2, resulting in negative current due to SCN^-^ outflow from the patch pipet [3,36,37,38]. When increasing the concentration of serine, anion currents increased in a concentration-dependent manner, saturating at high concentrations with an apparent *K*_m_ of 70 ± 18 μM (Figure 3D). In contrast, serine application to ASCT1-containing inside-out patches resulted in different behavior (Figure 3A), with substrate blocking the anion conductance, rather than activating it. Similar behavior of blocking the leak anion current was found for intracellular application of a competitive inhibitor (data not shown). From the serine concentration dependence, an apparent *K*_m_ value of 77 ± 34 μM was determined Figure 3B, when fit at low substrate concentrations. At concentrations exceeding 500 μM, a slight decrease in current was observed. The reason for this behavior is not known, although one possibility would be that ASCT1 has an allosteric binding site for the substrate. For ASCT2, these results indicate that substrate binds with slightly higher apparent affinity to the inward-facing conformation, whereas in ASCT1, apparent substrate affinities of the inward and outward-facing states are similar.

### 3.4. ASCT Activity Shows Biphasic Na^+^ Concentration Dependence

Evidence from molecular dynamics simulation and electrophysiology studies [1,2,5,8,39,40,41], points to the existence of three sodium coordination sites in ASCT1 and 2. By analyzing Na^+^ dependence of amino acid-induced anion currents, we determined the Na^+^ apparent affinity for both subtypes over a large [Na^+^] range, as shown in Figure 4. Total cation concentration was maintained at 140 mM by balancing with NMG^+^. As observed previously for ASCT2, apparent outward-directed currents were generated by applying amino acid substrate at very low Na^+^ concentration, ranging from 0.1 to 1 mM for both ASCT1 and 2 (Figure 4A,D), although the magnitude of the outward current was much smaller in ASCT1, presumably due to its lower leak anion conductance (see above). This apparent outward current is caused by a block of the tonic leak anion current (which is inward directed) by the amino acid substrate at low [Na^+^]. At higher Na^+^ concentrations, the current direction reversed to generate inward current (substrate-induced anion conductance (Figure 4B,E)). When plotted against the Na^+^ concentration (Figure 4C,F), a bi-phasic current-[Na^+^] relationship was observed, indicating that more than one Na^+^ binding step is required [5,8,21]. With fitting to a two-binding site equation, as shown by the red curves in Figure 4C,F, the following *K*_m_ values for Na^+^ binding were observed: hASCT2, *K*_Na1_ = 0.3 ± 0.1 mM, *K*_Na2_ = 27 ± 8 mM; hASCT1, *K*_Na1_ = 0.3 ± 0.5 mM, *K*_Na2_ = 14 ± 3 mM, indicating slightly weaker Na^+^ apparent affinity for the second cation binding steps in ASCT2 compared to ASCT1.

To further test the Na^+^ dependence of ASCT activity, we measured the substrate apparent affinities as a function of the Na^+^ concentration. As shown in Appendix A Appendix A, apparent substrate *K*_m_ values decrease with increasing extracellular Na^+^ concentration, as expected for cooperative binding of amino acid substrate and Na^+^ and as supported by the fits to a sequential Na^+^-substrate-Na^+^ binding model (solid lines in Appendix A Appendix A).

### 3.5. ASCT1 and 2 Have Comparable Kinetics of Substrate Translocation

The kinetics of reaction steps associated with ASCT2 substrate translocation were previously studied using rapid alanine application with laser photolysis of caged-alanine [5,8,34,42]. In this previous study, transient currents induced by alanine application decayed with a time constant of 3 ms, providing information about the electrogenic steps in the translocation equilibrium [20,21,22,43,44,45]. Here, we extended these experiments using a piezo-based rapid solution exchange device, allowing the rapid application, as well as removal of amino acid. These experiments were performed in the absence of permeating anions, to isolate electrogenic steps of the process. The intracellular solution contained Na^+^ and amino acid and, initially, the extracellular solution contained only Na^+^, conditions under which the amino acid binding site should be facing to the extracellular side of the membrane. Subsequently, serine was applied to the extracellular side of the membrane and later removed. As expected for electroneutral amino acid exchange, we did not observe any steady-state current. However, substantial transient inward current associated with cation binding and/or movement of the charged transport domain was caused by serine application [8,22] in both ASCT1 and 2, as shown in Figure 5A–D. The time constant for current decay was 4–5 ms after serine application. However, the kinetics of this decay could be partially limited by the rate of solution exchange, in particular for removal of the substrate. As expected, the transient currents were absent in the absence of extracellular Na^+^, suggesting that Na^+^ is a requirement for the electrogenic rearrangement of the ASCT binding sites within the membrane. Upon the removal of serine, transient current was observed in the opposite, outward direction, as expected for the re-equilibration of the translocation equilibrium to the outward-facing conformation, resulting in capacitive-like charge movement (i.e., the charge that moved inward during amino acid application has to move outward upon amino acid removal). Consistently, the outward charge movement upon substrate removal was of the same magnitude as the inward charge movement upon substrate application.

Next, we performed a paired-pulse experiment, where the time interval between removal of serine to a second application of serine was varied. This experiment can test how quickly the outward-facing conformation is restored after serine removal, providing information about the amino acid exchange turnover rate of ASCTs. As shown in Figure 6A,D, the peak serine-induced inward current recovered relatively rapidly after serine removal, with complete recovery after 50–100 ms for both transporters. We next performed a fit of the recovery amplitude with an exponential function. The rate constant for recovery was 60 ± 4 s^−1^ for ASCT1 (Figure 6B). and 75 ± 3 s^−1^ for ASCT2 (Figure 6E), suggesting that the recovery time constant is similar for both ASCT subtypes. Furthermore, we tested time constants at varying substrate concentrations, from 200 μM to 1000 μM, showing that the recovery rate remained unchanged (Figure 6C,F). This is expected because the re-equilibration of the exchange equilibrium should only depend on intracellular substrate concentration after removal of extracellular amino acid.

In addition, we tested kinetics associated with the substrate translocation step using different transported amino acids. As shown in Appendix A Appendix A, rate constants associated with recovery from serine and alanine application are in the same order of magnitude, but vary slightly for amino acid and subtype, with rate constants for ASCT2 being larger. Recovery rates after glutamine application are slower and no signal was observed for ASCT1, for which glutamine does not function as a transported substrate (Appendix A).

Finally, we determined the voltage dependence of substrate-induced transient currents, at membrane potentials varying from −100 mV to +60 mV (Appendix A Appendix A). Peak amplitudes of the transient current signal were plotted as a function of the membrane potential in Appendix A Appendix A. While voltage dependence of the peak current is weak for ASCT2, a stronger voltage dependence was observed for ASCT1, with current increasing about 1.07 fold for every −20 mV change in membrane potential (Appendix A Appendix A). As expected, the current increased at more negative potential, due to positive charge of the cotransported Na^+^ and/or loaded transport domain moving inward within the membrane. Overall, these results are consistent with a previous report [8], which did not find a strong voltage-dependent behavior of amino acid induced transient ASCT2 current.

## 4. Discussion

In this report, we compare the function of the neutral amino acid transporter members of the SLC1 family, ASCT1 and 2, using detailed electrophysiological and kinetic analysis. Despite minor differences in amino acid substrate specificity, with ASCT1 unable to recognize glutamine, the general functional properties are highly conserved between the two transporters. ASCT1 and 2 both catalyze a substrate-induced anion conductance, which is uncoupled with substrate transport. While both transporters also exhibit a leak anion conductance, in the absence of amino acid substrate, this leak conductance is much less pronounced in ASCT1 compared to ASCT2. Second, both transporter subtypes exhibit a biphasic response to Na^+^, suggesting that the sequential Na^+^-amino acid-Na^+^ binding mechanism is conserved. However, the apparent affinity of the Na^+^ binding step following substrate binding is somewhat lower in ASCT2 compared to ASCT1. Third, apparent affinities for the amino acid substrate serine binding to the inward-facing binding site are in the sub-100 μM range. However, serine is an inhibitor of the leak anion conductance in ASCT1 when applied to the intracellular side of the membrane, in contrast to ASCT2, where it is an activator. Finally, electrogenic transport steps in the translocation reaction are conserved in ASCT1 and 2 and occur with similar rates, suggesting that the turnover rates of the two transporter subtypes for amino acid exchange are > 65 s^−1^.

Amino acid and Na^+^ binding were proposed to occur in a sequential fashion, similar to the EAAT members of the family, with at least one Na^+^ binding to the *apo*-form of the transporter and another Na^+^ binding to the amino acid-bound form. This sequential binding sequence appears to be conserved in both ASCT subtypes. In addition, both transporters show the biphasic Na^+^ dose response relationship, indicating that the Na^+^ binding step to the empty transporter is of high affinity in both ASCT1 and 2, with an apparent *K*_m_ value < 1 mM. This suggests that this cation binding step is always saturated under physiological conditions, unless [Na^+^] would reach exceedingly low values. In contrast to ASCT2, the block of leak anion current by substrate at low sodium concentrations is not very pronounced, in agreement with the idea that the leak anion conductance is small in ASCT1. If there is a physiological significance for this difference in function is unknown. The [Na^+^] binding step following substrate binding is of lower affinity, in the 20 mM range, although this step is also close to being saturated at physiological [Na^+^]. This finding is in agreement with the hypothesis that Na^+^ plays a modulatory role in ASCTs, as it is not providing a driving force for amino acid exchange. Overall, these data are in excellent agreement with a recent report on ASCT2 showing a Na^+^/glutamine stoichiometry of 2:1 and, for the first time, directly demonstrating sodium flux across the membrane during amino acid transport [46].

The binding properties for amino acid from the intracellular side (inward-facing conformation) had not been previously characterized for ASCTs using electrophysiological analysis. However, using radiolabeled substrates uptake assay with sided proteoliposomes, Indiveri and colleagues were able to determine both the internal and external *K*_m_ for ASCT2 for several amino acids [32,47,48]. Interestingly, in their assay, serine had a 100-fold higher *K*_m_ in the inside-out proteoliposome configuration. In contrast, our data suggest that the internal and external *K*_m_ values are in the same 50–100 μM range. The reason for this discrepancy is not known. One possibility would be that Indiveri and colleagues reconstituted purified human ASCT2 in proteoliposomes, whereas our experiments were performed in HEK293 cells, which may contain regulatory factors that are absent in the purified transporter. Second, it is possible that the experimental *K*_m_ values differ for activation of the anion conductance and for actual substrate transport. In any case, the potential symmetry/asymmetry of amino acid affinities of the exchanger binding sites has strong implications on our understanding of ASCT function in the physiological environment and needs to be investigated more thoroughly. Another interesting result from the inside-out patch experiments is that serine inhibits the leak anion conductance, rather than activating an anion conductance, as is the case in ASCT2. 

In a previous report, it was shown that rapid amino acid application to ASCT2 resulted in inwardly-directed transient currents, most likely caused by the movement of charged Na^+^ into the transport domain, or by structural changes of the charged transport domain due to amino acid translocation, or both [5]. While rate constants for amino acid translocation were determined in this previous work, the turnover rate was only estimated. Here, we extended this previous analysis by using a paired-pulse amino acid application protocol, allowing the determination of the recovery rate of the current upon amino acid removal. This recovery rate is a better indicator of turnover rate than the rate constant of transient current after amino acid application, since the latter only informs on electrogenic partial reactions. The recovery rate was independent of extracellular serine concentration, as expected, but changed when the nature of the amino acid substrate was varied. For serine, a recovery rate constant of 75 s^−1^ was obtained for ASCT2 and 60 s^−1^ for ASCT1 (Figure 6), providing a lower limit for the turnover rate, since a slight limitation of these rate constants by solution exchange cannot be fully excluded. In any case, these values provide an estimate of amino acid exchange rates by these SLC1 family subtypes, suggesting that ASCT2 is a marginally faster exchanger than ASCT1. The knowledge of these turnover rates will have substantial implications for our understanding of the impact of ASCTs in physiological, as well as pathophysiological processes related to the metabolic turnover of neutral amino acids, in particular glutamine.

In summary, the work presented here compares the functional and kinetic properties of the neutral amino acid transporters ASCT1 and 2 from the SLC1 family. Both transporters share the high affinity for Na^+^ in the *apo*-, amino acid-free configuration, suggesting that this Na^+^ binding site (likely the Na3 site) is always occupied under physiological conditions. In contrast, the Na^+^ binding step following amino acid binding has a much lower apparent affinity, in the 15 to 25 mM range. Both transporters show sub-millimolar apparent affinities for serine in the inward-facing conformation, suggesting a lack of sidedness with respect to substrate affinities. Finally, despite overall electroneutral amino acid exchange, both transporters display fast electrogenic charge movement in response to rapid amino acid application, most likely caused by rearrangements of the transport domain during substrate translocation, or Na^+^ binding. From the recovery of these charge movements after amino acid removal, the turnover rate was estimated in the range of >65 s^−1^ for ASCT2 and only slightly slower for ASCT1. Together these data may be useful to understand the involvement of ASCTs in amino acid homeostasis under physiological conditions, as well as in certain cancer cells, in which ASCT2 is upregulated.

## Figures and Tables

**Figure 1 biomolecules-12-00113-f001:**
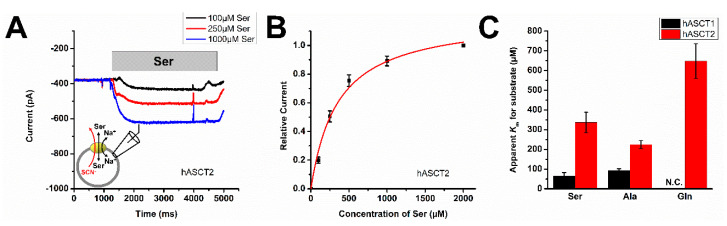
ASCT1 and 2 apparent substrate affinity and selectivity. (**A**) Typical current recording example with ASCT2 at varying serine concentrations (time of application indicated by the grey bar). (**B**) The serine apparent *K*_m_ was determined as 350 ± 60 μM for ASCT2 by fit of a Michaelis-Menten-like equation (red line). Relative current was determined by normalizing all current responses to the response obtained at 2 mM serine. (**C**) Comparison of *K*_m_ values for three substrates of ASCT1 and ASCT2. N.C. indicates no current was observed. Whole cell current recording experiments were performed under homo-exchange conditions: 130 mM NaSCN and 10 mM of serine in the pipette solution, the extracellular solution was 140 mM NaMes and variable concentration of substrate at V = 0 mV.

**Figure 2 biomolecules-12-00113-f002:**
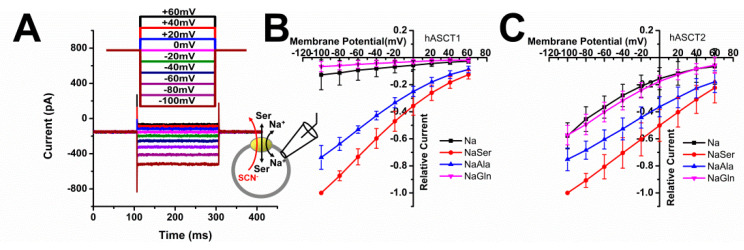
Voltage dependence of substrate-induced and leak anion currents. (**A**) Typical current recording in response to voltage jumps (from −100 to + 60 mV, in increments of 20 mV for the duration of 200 ms, voltage protocol shown at the top) from an ASCT1-transfected cell in the presence of extracellular NaMes and intracellular NaSCN with 10 mM serine. 400 μM (R)-gamma-(4-Biphenylmethyl)-L-proline was used to eliminate the unspecific background currents by subtraction. The anion current-voltage relationship at steady-state is plotted in (**B**) for ASCT1 and (**C**) for ASCT2.

**Figure 3 biomolecules-12-00113-f003:**
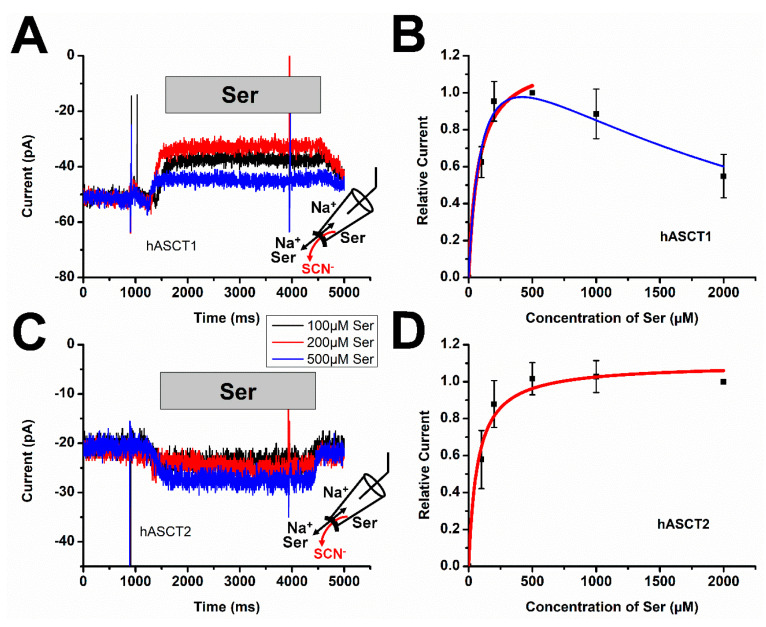
Determination of substrate apparent affinity of the inward-facing state. Representative whole cell current recordings from ASCT1 (**A**) and 2 (**C**) transfected cells in the inside-out patch configuration. The extracellular solution (pipet solution) contained 130 mM NaSCN with 10 mM Ser. The intracellular (bath) solution contained 140 mM NaMes with varying concentrations of serine. (**B**) Concentration dependence of serine-induced current for determining the apparent affinity for serine of the inward-facing state (*K*_m_ = 77 ± 34 μM with one site binding fitting equation, or *K*_1_ = 105 ± 12 μM, *K*_2_ = 2400 ± 200 μM for two binding site fitting for ASCT1). (**D**) Determination of apparent affinity of serine of the inward-facing state with a *K*_m_ of 70 ± 18 μM for ASCT2.

**Figure 4 biomolecules-12-00113-f004:**
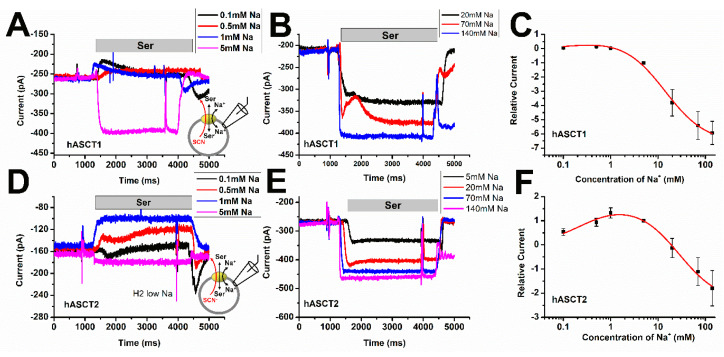
ASCT1 has slightly higher apparent affinity for Na^+^ association to the serine-bound transporter than ASCT2. Whole-cell current recordings were performed to test the apparent affinity for Na^+^ of ASCT transporters. (**A**,**D**) Typical serine-induced ASCT1 and 2 anion currents, measured under homo-exchange conditions at varying low [Na^+^] (0–5 mM). Serine concentration was saturating (10 mM) extracellularly, with 130 mM NaSCN and 10 mM Ser inside the pipet. The duration of serine application is indicated by the gray bar. (**B**,**E**) Similar experiments as in (**A**,**C**), but in a [Na^+^] range from 5 mM to 140 mM. (**C**,**F**) Serine induced maximum currents plotted as a function of [Na^+^]. The *K*_m_ for Na^+^ was determined from fitting dose-response curves of a two Na^+^ binding site function. *I* = *I*_max1_ * [Na^+^]/(*K*_Na1_ + [Na^+^]) + *I*_max2_ * [Na^+^]/(*K*_Na2_ + [Na^+^]). For ASCT2, *K*_Na1_ 0.3 ± 0.1 mM, *K*_Na2_ is 27.9 ± 8.8 mM. For ASCT1, *K*_Na1_ is 0.3 ± 0.1 mM, *K*_Na2_ is 14.0 ± 3.4 mM. In all experiments, the transmembrane potential was 0 mV.

**Figure 5 biomolecules-12-00113-f005:**
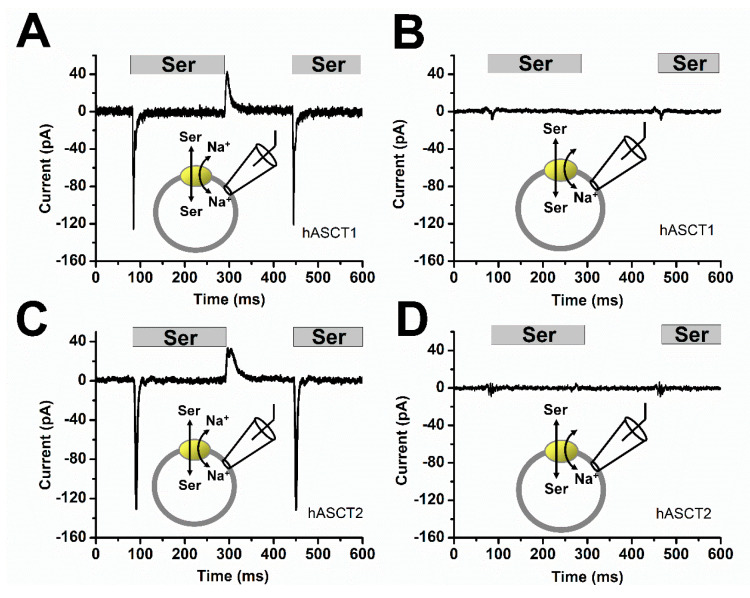
Rapid substrate application and removal induce capacitive-like transient currents in both ASCT1 and 2. (**A**,**C**) Typical current recording traces upon rapid solution exchange and removal using a θ tube. Experiments were performed with 140 mM Na^+^ together with 500 μM serine at the extracellular side, the intracellular solution contained 130 mM NaMes and 10 mM serine. Time constants for the decay of the transient current were τ_1_ = 2.3 ± 0.1 ms for serine application and τ_2_ = 7.8 ± 0.2 ms for serine removal (ASCT1) and the charge movements Q_1_ = 290 fC and Q_2_ = 340 fC, respectively. For ASCT2 the time constants were τ_1_ = 3.1 ± 0.1 ms and τ_2_ = 11.4 ± 0.3 ms. (**B**,**D**) Similar experiments but in the absence of external Na^+^.

**Figure 6 biomolecules-12-00113-f006:**
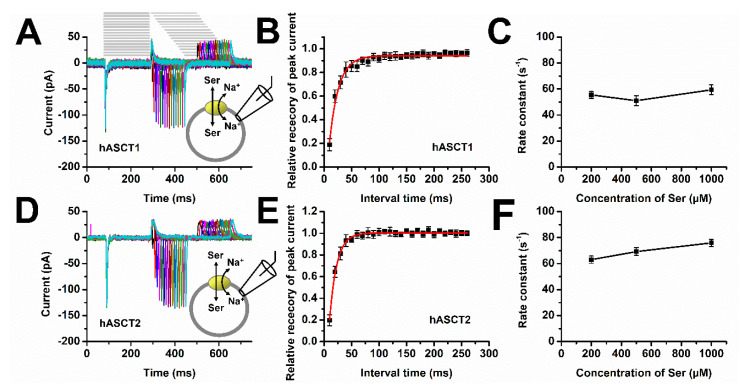
Paired-pulse experiment reveals recovery time constant after substrate removal. Paired-pulse experiments were performed with 140 mM NaMes and 200, 500, or 1000 μM serine in the extracellular solution and with 130 mM NaMes and 10 mM serine in the pipet solution. (**A**,**D**) Typical set of currents with an increase of recovery time of 10 ms in each trace (the bars at the top show the solution change protocol) at 500 μM serine. Peak amplitudes from (**A**,**B**) recovery currents were fitted using a single-exponential function (I = I_0_(1 − exp(−kt)) (**B**), ASCT1 and (**E**) ASCT2, yielding recovery rate constants of 60 ± 4 s^−1^ (**B**) and 75 ± 3 s^−1^ (**E**). The recovery rate constant was independent of the extracellular serine concentration for both ASCT1 (**C**) and 2 (**F**).

## Data Availability

Original data can be found in the Appendix A or will be made available upon request.

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
