# Peer review of "Functional and Kinetic Comparison of Alanine Cysteine Serine Transporters ASCT1 and ASCT2"

_biomolecules, 2022, doi:10.3390/biom12010113_

Round 1

Reviewer 1 Report

The manuscript by Wang et al is an interesting comparison of functional aspects of the two close transporters ASCT1 and 2. The manuscript is clearly written and sound, the experiments are well conceived. I have only minor suggestions for improving interpretation of some results.

Lines 136-137 (most experiments were performed with human 136 ASCT1 and 2, unless stated otherwise). The experiments in which non-human ASCTs have been used are not easily recognizable. Some speculation on specie specificity could be added in discussion section.

Line 192. The observed phenomenon could be inferred to substrate inhibition from internal side. Such a phenomenon is currently described for enzymes as due to binding of the substrate to an additional allosteric site, that could be the case of ASCT1.

Section 3.4 (biphasic Na+ concentration dependence). The observations described in this section are in good agreement with recent results on ASCT2/sodium interaction (Mazza et al FEBS Lett 2021). The correlation can confirm the author hypothesis.

Section 3.5. The conclusion of this elegant experiment is quite convincing. The same order of magnitude observed (line 260) shows indeed a ratio of about 3 out/in of absolute charge movement. May this be due to a preferential flux of sodium from outside to inside that, during the physiological transport, may result in a net inward flux of sodium associated to the amino acid antiport? This might correlate well with the cell gradient of sodium. Some comments could be added if authors agree with this view.

Author Response

Answers to Reviewers' Comments and Questions

We would like to thank both reviewers for their detailed reading of our manuscript and their thoughtful comments and questions.  Below please find our point-by-point answer to these questions and comments.

Reviewer 1:

"Lines 136-137 (most experiments were performed with human 136 ASCT1 and 2, unless stated otherwise). The experiments in which non-human ASCTs have been used are not easily recognizable. Some speculation on specie specificity could be added in discussion section."  Answer:  The only experiments done with rASCT2 were in a previous version of the manuscript, but had later been removed.  Therefore, this sentence was removed.

"Line 192. The observed phenomenon could be inferred to substrate inhibition from internal side. Such a phenomenon is currently described for enzymes as due to binding of the substrate to an additional allosteric site, that could be the case of ASCT1."  Answer:  Yes, this is a very interesting possibility, which we had taken into account, but we didn't want to speculate too much.  We now mention this as a possible explanation of the biphasic dose response behavior. 

"Section 3.4 (biphasic Na+ concentration dependence). The observations described in this section are in good agreement with recent results on ASCT2/sodium interaction (Mazza et al FEBS Lett 2021). The correlation can confirm the author hypothesis.Answer:  Thanks for this observation.  We now cite and briefly discuss the Mazza et al. 2021 paper, but rather in the discussion section than in the results section. 

"Section 3.5. The conclusion of this elegant experiment is quite convincing. The same order of magnitude observed (line 260) shows indeed a ratio of about 3 out/in of absolute charge movement. May this be due to a preferential flux of sodium from outside to inside that, during the physiological transport, may result in a net inward flux of sodium associated to the amino acid antiport? This might correlate well with the cell gradient of sodium. Some comments could be added if authors agree with this view."  Answer:  While the ratio of the amplitudes of the on/off transient currents is roughly 3:1, the off current also decays with a roughly 3-times larger time constant.  Therefore, the area under the on and off curves (the total charge moved) is about the same.  We now give the exact values for the moved charge in fC for both on and off currents to clarify this point.

Reviewer 2 Report

This is a very nice study on the functional and kinetic differences of two subtypes of neutral amino acid transporters (ASCT1 and ASCT2)

I have only a few comments.

1.) It would be nice to have an insert in Fig.1A of a recording, which shows that at a zero-trans condition there is no current induction by the application of substrate to the cell.

2.) The results from the inside out recordings are interesting but confusing. It is not clear to me why it was not tried to define the leak current with an inhibitor.  ASCT1  carried a leak current at rest, which was blocked by the substrate. Is there also a leak current in ASCT2 that increases when substrate is applied? Such data could provide information on where in the transport cycle the conducting state resides. However, if the authors think that this is subject of a future investigation I am fine with that.

3.) I do not understand why there is an outward current at low Na+concentrations? It seems that in the experiment SCN is only present in the pipette solution. Hence, it can leave the cell (inward current) but not enter it (outward current).  Either SCN was also in the outer solution, but then this should be indicated or another anion entered. If the latter is the case please tell which one it is.

4.) Please describe the procedure for obtaining "Relative Current"

5.) Please prepare a Fig.6 instead of referring to data in the supplement

Author Response

Answers to Reviewers' Comments and Questions

We would like to thank both reviewers for their detailed reading of our manuscript and their thoughtful comments and questions.  Below please find our point-by-point answer to these questions and comments.

Reviewer 2:

"This is a very nice study on the functional and kinetic differences of two subtypes of neutral amino acid transporters (ASCT1 and ASCT2)Answer:  Thank you for the kind remarks.

"I have only a few comments.

1.) It would be nice to have an insert in Fig.1A of a recording, which shows that at a zero-trans condition there is no current induction by the application of substrate to the cell."  Answer:  Under zero-trans conditions, i.e. no permeable anion inside the cell and same amino acid/sodium concentrations on both sides of the membrane, the relevant currents are shown in Fig. 5, after the initial transient current response has decayed (note the absence of steady state current).  We now added a remark about this in the description of Fig. 1.

"2.) The results from the inside out recordings are interesting but confusing. It is not clear to me why it was not tried to define the leak current with an inhibitor.  ASCT1  carried a leak current at rest, which was blocked by the substrate. Is there also a leak current in ASCT2 that increases when substrate is applied? Such data could provide information on where in the transport cycle the conducting state resides. However, if the authors think that this is subject of a future investigation I am fine with that."  Answer: Leak current is blocked by the amino acid substrate, and also by competitive inhibitors (not shown here). In contrast to amino acid substrates, which block leak current at low [Na+] only (they activate anion current at high [Na+]), competitive inhibitors block leak anion current at any [Na+].  While we do not want to show extensive data on this aspect in this manuscript, we now mention it in this section.      

"3.) I do not understand why there is an outward current at low Na+ concentrations? It seems that in the experiment SCN is only present in the pipette solution. Hence, it can leave the cell (inward current) but not enter it (outward current).  Either SCN was also in the outer solution, but then this should be indicated or another anion entered. If the latter is the case please tell which one it is."  Answer:  The apparent outward current at low [Na+] is in fact the block of a tonic inward anion current (leak current) that is activated by Na+ and is already present in the absence of amino acid substrate.  We have revised this paragraph accordingly, to make this point more clear.

"4.) Please describe the procedure for obtaining "Relative Current"Answer:  This is now explained in the legend of Fig. 1.

"5.) Please prepare a Fig.6 instead of referring to data in the supplementAnswer:  Since we provide three supplementary figures, we feel that the manuscript would be too long if all of these figures were included in the main manuscript.  Therefore, we would prefer to keep the supplementary figures file as is.